# Round complexity in the local transformations of quantum and classical states

Eric Chitambar[1] & Min-Hsiu Hsieh[2]

In distributed quantum and classical information processing, spatially separated parties operate locally on their respective subsystems, but coordinate their actions through multiple exchanges of public communication. With interaction, the parties can perform more tasks. But how the exact number and order of exchanges enhances their operational capabilities is not well understood. Here we consider the minimum number of communication rounds needed to perform the locality-constrained tasks of entanglement transformation and its classical analog of secrecy manipulation. We provide an explicit construction of both quantum and classical state transformations which, for any given $r$, can be achieved using $r$ rounds of classical communication exchanges, but no fewer. To show this, we build on the common structure underlying both resource theories of quantum entanglement and classical secret key. Our results reveal that highly complex communication protocols are indeed necessary to fully harness the information-theoretic resources contained in general quantum and classical states.

[1] Department of Physics and Astronomy, Southern Illinois University, Carbondale, IL 62901, USA. [2] Centre for Quantum Software and Information (CQSI), Faculty of Engineering and Information Technology (FEIT), University of Technology Sydney (UTS), Sydney, NSW 2007, Australia. Correspondence and requests for materials should be addressed to E.C. (email: echitamb@siu.edu) or to M.-H.H. (email: minhsiuh@gmail.com)

O ne of the most fascinating aspects of quantum information is how classical communication can enhance quantum information processing. For instance, "mixed" entanglement shared between two or more parties can be "purified" when the parties are allowed to perform Local quantum Operations on their subsystems and Communicate Classically with one another[1], a process known as LOCC. An analog of this purification procedure can be found in the classical theory of secret correlations. Rather surprisingly, secret correlations shared between two or more parties can be strengthened by the parties performing Local stochastic Operations and "leaking" information partially through Public Communication[2–4], a process known as LOPC. In all LOCC/LOPC protocols, the parties take turns exchanging information with one another, information that is extracted locally from their respective subsystems and earlier rounds of communication. In both settings, the principle is the same: resource manipulation (whether it be entanglement or secrecy) becomes more powerful when public classical communication is allowed.

For a distributed information-theoretic task, its communication complexity quantifies the minimum amount of messages that must be exchanged in order to perform the task[5–8]. A more fine-grained notion of communication complexity emerges by considering communication exchanges within a certain number of rounds. The $r$-round communication complexity of a task is the minimum amount of communication needed to perform the task in a protocol, lasting no more than $r$-rounds. The subject of $r$-round communication complexity has received notable attention on both the classical and quantum sides[9–14]. One of the most well-known examples is the so-called "pointer jumping" problem, which is known to demonstrate an exponential gap between the $r$ − 1 and $r$-round communication complexity when computing the $r^{\text{th}}$ pointer value[15–17]. Even stronger, such a gap exists if quantum communication is allowed[12] (i.e., the parties can exchange qubits with one another each round, something prohibited in the LOCC model).

The complexity of a distributed information-processing task can also be measured in terms of its round complexity, which quantifies the minimum number of communication rounds needed to perform the task, regardless of the total communication cost. Round complexity becomes an important question in distributed tasks where the parties do not want to share with one another all their local information, such as in secure function computation[18] and interactive prover scenarios[19]. Round complexity is also a meaningful measure of complexity when constraints are placed on the allowed types of communication, particularly in the LOPC/LOCC frameworks where private/quantum communication is not allowed. Without such constraints, one round of communication is always sufficient to perform any distributed information processing task since one party can just transmit all of his/her local information to the other. In contrast, certain tasks such as generating secret correlations/quantum entanglement are impossible in the LOPC/LOCC settings, even when the parties can exchange an unbounded amount of public communication.

LOPC/LOCC round complexity is thus a fundamental property of multi-party tasks that captures the necessity of interaction for optimal resource manipulation. An important practical motivation for studying this property follows from relativistic constraints that place a limitation on how quickly messages can be sent between spatially separated parties. Round complexity thereby places a lower bound on the time needed to perform a given task. As a result, with more round complexity, there is a greater reliance on classical or quantum memories, something generally undesirable due to the fragility of quantum systems.

A comprehensive understanding of LOCC/LOPC round complexity is still lacking. It is known that two-round protocols can be strictly superior to one-way schemes for distilling classical key from wire-tapped sources[3]. On the quantum side, clear separations between one-round and two-round protocols have been demonstrated for various quantum information-processing tasks such as asymptotic entanglement distillation[1], tripartite entanglement transformations[20], quantum state discrimination[21–25], and, recently, the simulation of nonlocal gates using shared entanglement[26]. However, none of these results have been able to establish an operational separation between each of the finite-round LOCC classes in terms of the most basic LOCC tasks: manipulating a quantum state from one form to another. It has been previously unknown whether or not every LOCC transformation $\rho \rightarrow \sigma$ can be completed using a constant number of communication rounds, even if an unbounded amount of classical communication is allowed.

Here we construct, for any integer $r \geq 1$ families of quantum (resp. classical) states for which a minimum of $r$ communication rounds is both necessary and sufficient to obtain pure-state entanglement (resp. secret shared randomness). Such a phenomenon might be unexpected given that every bipartite pure-state transformation $|\psi\rangle \rightarrow |\phi\rangle$ can be accomplished in just one round of LOCC, regardless of the dimensions[27]. Our findings imply that there exists no universal upper bound on the number of LOCC/LOPC rounds needed to perform such tasks, universal in the sense that it holds for states of all dimensions/alphabet size. Rigorously proving that this claim is a delicate matter since the general structure of LOCC and LOPC protocols is quite complex, allowing for arbitrary local operations and arbitrary interactive communication schemes[28]. With this complexity, it is difficult to definitively rule out the possibility of some clever round-compression technique that could always reduce the number of communication exchanges below some finite threshold, regardless of the system sizes. In fact, such a clever round-compression strategy is precisely what allows for the restriction to just one-way protocols for all bipartite pure-state transformations $|\psi\rangle \rightarrow |\phi\rangle$[29]. Our work makes use of an analogous structure underlying both LOCC and LOPC protocols, and it conducts an analysis that applies to both the quantum and classical problems.

## Results

**The LOCC and LOPC frameworks**. The problems studied in this paper involve two trustworthy parties (Alice and Bob) and one unwanted third party (Eve). When Alice, Bob, and Eve are holding quantum systems, we denote their joint state by $\rho^{ABE}$. In contrast when Alice, Bob, and Eve are holding random variables $X$, $Y$, and $Z$, we denote their joint probability distribution by $p^{XYZ}$. These variables range over sets $\mathcal{X}$, $\mathcal{Y}$, and $\mathcal{Z}$, respectively, and the probability of event $(x, y, z)$ will be denoted by $p_{xyz}^{XYZ}$. When the underlying random variables are clear, we will simply write the probabilities by $p_{xyz}$. Conditional probabilities are denoted, for example, by $p_{xy|z}^{XY|Z=z}$.

In an LOCC protocol, Alice and Bob take turns performing a local quantum instrument, which is a collection of completely positive (CP) maps $\{\mathcal{E}_\lambda\}_\lambda$ such that $\sum_\lambda \mathcal{E}_\lambda$ is trace-preserving[28]. The index $\lambda$ represents the "measurement outcome" of the instrument, which is communicated to the other party, thereby correlating the choice of future local instruments to previous measurement outcomes. For the problem considered in this paper, we will be considering instruments in which each local CP maps has the form $\mathcal{E}_\lambda(\rho) = K_\lambda \rho K_\lambda^\dagger$, where the $\{K_\lambda\}_\lambda$ form a complete set of Kraus operators; i.e., $\sum_\lambda K_\lambda^\dagger K_\lambda = \mathbb{I}$.

In an LOPC protocol, Alice and Bob share random variables $X$ and $Y$, respectively. They proceed with multiple iterations of

public communication where the $i^{th}$ message $M_i$ is the stochastic output of a channel performed to $(P, M_{<i})$, where $P \in \{X, Y\}$ is the variable of the announcing party in the $i^{th}$ round and $M_{<i} = M_1 \cdots M_{i-1}$ denotes the sequence of messages generated in the previous $i-1$ rounds. At the end of the protocol, Alice and Bob generate output variables $\hat{X}$ and $\hat{Y}$ that are obtained by processing $(X, M)$ and $(Y, M)$, respectively, where $M$ represents all communication variables generated throughout the protocol. For both LOCC and LOPC, an $r$-round protocol consists of $r$ classical communication exchanges between the parties.

One conceptual difference between the LOCC and LOPC settings is that in the latter, the presence of an unwanted eavesdropping party is always taken into account. Thus, a copy of the public communication $M$ is shared by Eve, and a general LOPC protocol generates a transformation of probability distributions

$$p^{XYZ} \rightarrow p^{\hat{X}\hat{Y}(ZM)}. \tag{1}$$

The fundamental resource unit in entanglement theory is the entangled bit (ebit), which has the form $|\Phi\rangle^{AB} = \sqrt{1/2}\left(|00\rangle^{AB} + |11\rangle^{AB}\right)$. In classical secrecy theory, the basic resource unit is the secret bit (sbit). This is any distribution over the sets $\{0, 1\} \times \{0, 1\} \times \mathcal{Z}$ of the form $p_{xyz}^{XYZ} = \frac{1}{2}\delta_{xy}p_z^Z$, where $p^Z$ is an arbitrary distribution for Eve. Alice and Bob's main concern is how much Eve is correlated with their variables, rather than the specific distribution over her variable. Hence, we will adopt the notation that $\Phi$ denotes a sbit, with Eve's uncorrelated distribution being unspecified. For partially entangled two-qubit states and for non-uniform secret-shared bits, we will write

$$|\Phi_\lambda\rangle = \sqrt{\lambda}|00\rangle + \sqrt{1-\lambda}|11\rangle \tag{2}$$

$$\Phi_\lambda = \lambda\delta_{X0}\delta_{Y0} + (1-\lambda)\delta_{X1}\delta_{Y1}. \tag{3}$$

Here $\delta_{X0}$, for example, is the distribution over $\mathcal{X}$ that has $x = 0$ with unit probability. The entropy of $\Phi_\lambda$ is $h(\lambda)$, where $h(x) = -x\log x - (1-x)\log(1-x)$.

**Transformations with high round complexity.** Our results are based on a family of tripartite distributions, which we call the origami distributions. The family is given by the set $\left\{ \mathbf{b}^{(i,\lambda)} : i \in \mathbb{N}, 0 < \lambda \leq 1/2 \right\}$, with $\mathbf{b}^{(i,\lambda)}$ being a tripartite probability distribution taking on values $\mathbf{b}_{xyz}^{(i,\lambda)}$ for each fixed pair of values $(i, \lambda)$; i.e., event $(x, y, z)$ occurs with probability $\mathbf{b}_{xyz}^{(i,\lambda)}$ and $\sum_{xyz} \mathbf{b}_{xyz}^{(i,\lambda)} = 1$. The structure of these distributions is described recursively with $\mathbf{b}^{(1,\lambda)}$ having the form:

$$\mathbf{b}^{(1,\lambda)} = \begin{array}{c} \\ \\ Y \end{array} \begin{array}{c|cccc} & \multicolumn{4}{c}{X} \\ & 0 & 1 & 2 & 3 \\ \hline 0 & 0 & \cdot & \cdot & 1 \\ 1 & \cdot & 0 & 1 & \cdot \\ 2 & 2 & 3 & \cdot & \cdot \\ 3 & \cdot & \cdot & 3 & 2 \end{array} \begin{array}{c} \\ \\ \\ \\ Z \end{array} \tag{4}$$

The 8 events of $(x, y, z)$ having nonzero probability in $\mathbf{b}^{(1,\lambda)}$ are those in which $z$ lies in row $y$ and column $x$, as shown in this grid. The probabilities of these events are $\mathbf{b}_{xy|z}^{(1,\lambda)} = \lambda$ for even values of $x$, $\mathbf{b}_{xy|z}^{(1,\lambda)} = 1 - \lambda$ for odd values of $x$, and $\mathbf{b}_z^{(1,\lambda)} = 1/4$. In other words, $\mathbf{b}^{(1,\lambda)}$ consists of four blocks of uniform probability, each corresponding to a different value of $z$. Within each block, $X$ and $Y$ are perfectly correlated, but with a nonuniform distribution $(\lambda, 1-\lambda)$. For example, the event $(X = 1, Y = 2, Z = 3)$ occurs with

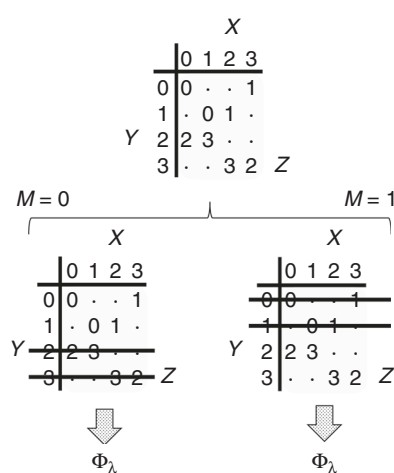

**Fig. 1** A One-Round LOPC for Obtaining Secret Correlations from $\mathbf{b}^{(1,\lambda)}$. Bob announces whether the value of $Y$ belongs to either $\{0, 1\}$ or $\{2, 3\}$. Using this information, Alice can determine the values of both $Y$ and $Z$ from her value of $X$. Relabeling then brings Alice and Bob's distribution into standard form $\Phi_\lambda$, independent of $Z$

probability 1/8 according to distribution $\mathbf{b}^{(1,\lambda)}$, while the event $(X = 1, Y = 2, Z = 2)$ occurs with probability zero.

Before describing the full set of distributions $\{\mathbf{b}^{(i,\lambda)}\}_{i\in\mathbb{N}}$, let us note the crucial structural properties of $\mathbf{b}^{(1,\lambda)}$ that we will want to generalize. The distribution $\mathbf{b}^{(1,\lambda)}$ decomposes into two parts — the first with $Y \in \{0, 1\}$ and the second with $Y \in \{2, 3\}$—such that within each of these sub-parts Alice can determine both the value of $Y$ and $Z$ from her value of $X$ (see Fig. 1). A simple one-round protocol allowing Alice to "unlock" this information involves Bob announcing message $M$ where $M = 0$ if $Y \in \{0, 1\}$ and $M = 1$ if $Y \in \{2, 3\}$. This message will not leak any information to Eve, since by the structure of $\mathbf{b}^{(1,\lambda)}$, $M$ is also a function of $Z$; i.e., Eve learns no more information about $XY$ from $M$ than what she already knows from $Z$. Secret correlations $\Phi_\lambda$ are then obtained by Alice and Bob applying the local functions:

$$\hat{X} = \begin{cases} 0 \text{ if } X \in \{0, 3\} \text{ and } M = 0, \text{ or } X \in \{1, 2\} \text{ and } M = 1 \\ 1 \text{ if } X \in \{0, 1\} \text{ and } M = 1, \text{ or } X \in \{2, 3\} \text{ and } M = 0 \end{cases},$$

$$\hat{Y} = \begin{cases} 0, \text{ if } Y \text{ is even} \\ 1, \text{ if } Y \text{ is odd} \end{cases}. \tag{5}$$

Even in this relatively simple case, it is not immediately obvious that the described LOPC protocol is essentially the only one-round protocol allowing Alice and Bob to establish secret key. However, the structure of $\mathbf{b}^{(1,\lambda)}$ ensures that Eve will have side information of any common function $J = \hat{X} = \hat{Y}$ computed locally by Alice and Bob unless at least one of the parties can determine the value of $Z$ (see Supplementary Note 1). The only way to achieve this in one-round of communication is by Bob revealing whether $Y \in \{0, 1\}$ of $Y \in \{2, 3\}$.

To build larger-round distributions, the idea will be to "copy and shift" $\mathbf{b}^{(1,\lambda)}$ into a multi-layered structure so that going from one layer to the next requires performing the one-round protocol just described. More precisely, for each fixed value of $\lambda$ the $i^{th}$ distribution in the family $\left\{ \mathbf{b}^{(i,\lambda)} : i \in \mathbb{N}, 0 < \lambda \leq 1/2 \right\}$ is built according to the following prescription:

$$\begin{cases} \text{Define:} \quad \overline{\mathbf{b}^{(1,\lambda)}} = \begin{array}{c} X \\ \begin{array}{c|cccc} & 0 & 1 & 2 & 3 \\ \hline 0 & 4 & \cdot & \cdot & 5 \\ 1 & \cdot & \cdot & 7 & 6 \\ 2 & 6 & 7 & \cdot & \cdot \\ 3 & \cdot & 4 & 5 & \cdot \end{array} \end{array} \; Y \qquad Z, \\ \\ \text{Even } n: \quad \mathbf{b}^{(n,\lambda)} = \left[ \mathbf{b}^{(n-1,\lambda)} \;\; \overline{\mathbf{b}^{(n-1,\lambda)}} \right] \qquad (\text{size: } 2^{n/2+2} \times 2^{n/2+1} \times 2^{n+1}), \\ \\ \text{Odd } n: \quad \mathbf{b}^{(n,\lambda)} = \left[ \begin{array}{c} \mathbf{b}^{(n-1,\lambda)} \\ \overline{\mathbf{b}^{(n-1,\lambda)}} \end{array} \right] \qquad (\text{size: } 2^{(n-1)/2+2} \times 2^{(n-1)/2+2} \times 2^{n+1}), \end{cases}$$
(6)

where $\overline{\mathbf{b}^{(n,\lambda)}}$ is obtained from $\mathbf{b}^{(n,\lambda)}$ by interchanging the row (resp. column) $i$ with row (resp. column) $i + 2^{\lfloor n/2+1 \rfloor}$ for all odd $i$ whenever $n$ is odd (resp. even), and Eve's values are increased by $2^n$ from the original values in $\mathbf{b}^{(n,\lambda)}$. In each grid, all of Eve's values are still equiprobable, and for each value of $z$, Alice and Bob have shared randomness with $\mathbf{b}^{(n,\lambda)}_{xy|z} = \lambda$ for even values of $x$. The distribution $\mathbf{b}^{(n,\lambda)}$ thus ranges over sets $\mathcal{X}$, $\mathcal{Y}$, and $\mathcal{Z}$ with respective cardinalities $|\mathcal{X}| = 2^{\lfloor n/2+2 \rfloor}$, $|\mathcal{Y}| = 2^{\lfloor n/2+3/2 \rfloor}$, and $|\mathcal{Z}| = 2^{n+2}$.

We call these origami distributions, due to the "unfolding" appearance of the construction. For example,

$$\mathbf{b}^{(2,\lambda)} = \begin{array}{c} X \\ \begin{array}{c|cccccccc} & 0 & 1 & 2 & 3 & 4 & 5 & 6 & 7 \\ \hline 0 & 0 & \cdot & \cdot & 1 & 4 & \cdot & \cdot & 5 \\ 1 & \cdot & 0 & 1 & \cdot & \cdot & 7 & 6 \\ 2 & 2 & 3 & \cdot & \cdot & 6 & 7 & \cdot & \cdot \\ 3 & \cdot & \cdot & 3 & 2 & \cdot & 4 & 5 & \cdot \end{array} \end{array} \; Y \qquad Z$$

$$\mathbf{b}^{(3,\lambda)} = \begin{array}{c} X \\ \begin{array}{c|cccccccc} & 0 & 1 & 2 & 3 & 4 & 5 & 6 & 7 \\ \hline 0 & 0 & \cdot & \cdot & 1 & 4 & \cdot & \cdot & 5 \\ 1 & \cdot & 0 & 1 & \cdot & \cdot & 7 & 6 \\ 2 & 2 & 3 & \cdot & \cdot & 6 & 7 & \cdot \\ 3 & \cdot & \cdot & 3 & 2 & \cdot & 4 & 5 & \cdot \\ 4 & 8 & \cdot & \cdot & 13 & 12 & \cdot & \cdot & 9 \\ 5 & \cdot & 9 & 14 & \cdot & \cdot & 8 & 15 & \cdot \\ 6 & 10 & 15 & \cdot & \cdot & 14 & 11 & \cdot & \cdot \\ 7 & \cdot & 12 & 11 & \cdot & \cdot & 13 & 10 \end{array} \end{array} \; Y \qquad Z$$
(7)

$$\mathbf{b}^{(4,\lambda)} = \begin{array}{c} X \\ \begin{array}{c|cccccccc|cccccccc} & 0 & 1 & 2 & 3 & 4 & 5 & 6 & 7 & 8 & 9 & 10 & 11 & 12 & 13 & 14 & 15 \\ \hline 0 & 0 & \cdot & \cdot & 1 & 4 & \cdot & \cdot & 5 & 16 & \cdot & \cdot & 17 & 20 & \cdot & \cdot & 21 \\ 1 & \cdot & 0 & 1 & \cdot & \cdot & 7 & 6 & \cdot & \cdot & 25 & 30 & \cdot & \cdot & 24 & 31 & \cdot \\ 2 & 2 & 3 & \cdot & \cdot & 6 & 7 & \cdot & \cdot & 18 & 19 & \cdot & \cdot & 22 & 23 & \cdot \\ 3 & \cdot & \cdot & 3 & 2 & \cdot & 4 & 5 & \cdot & \cdot & 28 & 27 & \cdot & \cdot & 29 & 26 \\ 4 & 8 & \cdot & \cdot & 13 & 12 & \cdot & \cdot & 9 & 24 & \cdot & \cdot & 29 & 28 & \cdot & \cdot & 25 \\ 5 & \cdot & 9 & 14 & \cdot & \cdot & 8 & 15 & \cdot & \cdot & 16 & 17 & \cdot & \cdot & 23 & 22 \\ 6 & 10 & 15 & \cdot & \cdot & 6 & 11 & \cdot & \cdot & 26 & 31 & \cdot & \cdot & 30 & 27 & \cdot \\ 7 & \cdot & 12 & 11 & \cdot & \cdot & 13 & 10 & \cdot & \cdot & 19 & 18 & \cdot & \cdot & 20 & 21 & \cdot \end{array} \end{array} \; Y \qquad Z.$$
(8)

We now use the origami distributions to construct bipartite quantum states. This is accomplished by first embedding each distribution $\mathbf{b}^{(i,\lambda)}$ into a tripartite quantum state according to

$$\left| \mathbf{b}^{(i,\lambda)} \right\rangle^{ABE} = \sum_{x,y,z} \sqrt{\mathbf{b}^{(i,\lambda)}_{xyz}} |x\rangle^A |y\rangle^B |z\rangle^E$$
$$= \frac{1}{\sqrt{2^{i+1}}} \sum_z \left| \psi_z^{(i,\lambda)} \right\rangle^{AB} |z\rangle^E,$$
(9)

where $\left| \psi_z^{(i,\lambda)} \right\rangle^{AB} := \sum_{x,y} \sqrt{\mathbf{b}^{(i,\lambda)}_{xy|z}} |x\rangle^A |y\rangle^B$. Notice that the von Neumann entropy of the reduced-state of $\left| \psi_z^{(i,\lambda)} \right\rangle^{AB}$ is $h(\lambda)$ for every $z$ and $i$. Alice and Bob's reduced state is then given by $\rho_{\mathbf{b}}^{(i,\lambda)} := tr_E \left( \left| \mathbf{b}^{(i,\lambda)} \right\rangle \left\langle \mathbf{b}^{(i,\lambda)} \right| \right)$. The main results of this paper are stated in the following theorem.

**Theorem 1.** For any pair $(r, \lambda)$ and any $0 < \lambda' \leq 1/2$, the LOPC transformation

$$\mathbf{b}^{(r,\lambda)} \rightarrow \Phi_{\lambda'}$$
(10)

and the LOCC transformation

$$\rho_{\mathbf{b}}^{(r,\lambda)} \rightarrow |\Phi_{\lambda'}\rangle$$
(11)

are both impossible using $r - 1$ rounds of communication exchanges, nor are they possible in $r$ rounds if Alice (resp. Bob) is

the first to announce when $r$ is odd (resp. even). Conversely, for $\lambda' \leq \lambda \leq 1/2$ the transformations are possible in $r$ rounds if Bob (resp. Alice) is the first to announce when $r$ is odd (resp. even).

**The classical secrecy rank.** A key tool in proving the classical part of Theorem 1 is, what we will call, the secrecy rank of a tripartite distribution. Its construction is based on the so-called secret key cost of a tripartite distribution[30], a quantity whose single-letter characterization[31,32] has close connections to Wyner's classic notion of common information[33]. A detailed exploration of the relationship between all these quantities is beyond the scope of this paper and will be saved for future work.

In what follows, for a distribution $p^W$ over the set $\mathcal{W}$, we let $|p^W|$ denote the number of events in $\mathcal{W}$ with a nonzero probability.

**Definition 1.** The secrecy rank of tripartite distribution $p^{XYZ}$ is defined as

$$\text{Srk}\left[ p^{XYZ} \right] = \min_{X-ZW-Y} \max_z \left| p^{W|Z=z} \right|,$$
(12)

where the minimization is taken over all auxiliary random variables $W$ such that $I(X:Y|ZW) = 0$.

Let us describe how this quantity is analogous to the quantum Schmidt rank. First consider the case when $Z$ is trivial; i.e., $|p^Z| = 1$. The Schmidt decomposition of a bipartite pure state has the form $|\varphi\rangle^{AB} = \sum_{w=1}^{\text{Srk}(|\varphi\rangle)} \sqrt{p_w} |\alpha_w\rangle^A |\beta_w\rangle^B$, where the $\{|\alpha_w\rangle^A\}$ and $\{|\beta_w\rangle^B\}$ form orthonormal bases for Alice and Bob's systems, respectively. Suppose that Alice and Bob both measure $|\varphi\rangle^{AB}$ by projecting in their Schmidt basis. If $X$ (resp. $Y$) is the random variable describing Alice's (resp. Bob's) outcomes, then their measurement statistics can be described as the marginal of the tripartite distribution $p^{XYW}$ where $p_{xyw}^{XYW} = \delta_{xw} \delta_{yw} p_w^W$; i.e., $X - W - Y$. Clearly $\text{Srk}(|\varphi\rangle) = \text{Srk}(p^{XY})$. In the case that $Z$ is not trivial, the definition of Eq. (12) most closely resembles the definition of Schmidt rank for bipartite mixed states, as proposed in ref. [34]. Namely, for a density matrix $\rho^{AB}$, one minimizes the quantity Srk ($\mathfrak{E}$) over all pure-state ensembles $\mathfrak{E} = \{|\psi_i\rangle^{AB}, q_i\}$ generating $\rho^{AB}$, where $\text{Srk}(\mathfrak{E})$ is the maximum Schmidt rank of all the states in $\mathfrak{E}$. For classical distributions $p^{XYZ}$, one can think of $p^{XYZ}$ as defining an ensemble of bipartite classical states $\mathfrak{E} = \{p^{XY|Z=z}, p_z^Z\}$. There is no minimization over ensembles as in the quantum case and therefore, one obtains the secrecy rank of $p^{XYZ}$ by just taking the maximum secrecy rank of all the states in $\mathfrak{E}$. This is precisely what Eq. (12) gives.

Another similarity between the secrecy rank and the Schmidt rank concerns their monotonicity.

**Theorem 2.** The secrecy rank is a stochastic LOPC (SLOPC) monotone.

Being an SLOPC monotone means that if the secrecy rank decreases at any point along any branch in an LOPC protocol, then it cannot be increased again along that branch. As described above, this property of the secrecy rank is the key ingredient in proving Theorem 1. The proof of Theorem 2 is given in the Supplementary Note 4.

**Discussion**

The operational tasks explored in this paper involve extracting pure-state entanglement from some mixed quantum state using LOCC and the classical analog of extracting secret shared randomness from an unsecure classically correlated state using LOPC. These are two very important questions since pure-state entanglement is the fundamental building block for quantum

information processing[35], and likewise, secret key states provide the essential ingredient for information-theoretic secure communication[36,37]. The tasks studied here can be seen as single-copy versions of the well-studied secret key[3,4] and entanglement distillation[38] problems. Understanding the relationship between quantum entanglement and classical secrecy offers an intriguing research directions with many interesting connections already found[39–48]. We have shown another similarity between the two in terms of LOCC/LOPC round complexity. Specifically, our results imply that no universal upper bound exists on the minimum number of rounds needed to optimally transform bipartite entanglement or extract secret shared randomness from unsecure correlations. We close this paper with some additional observations and open questions.

First, it should be emphasized that the LOCC impossibiltiy result of Theorem 1 holds for any $0 < \lambda' \leq \lambda$. In particular, the target state $|\Phi_{\lambda'}\rangle$ can be entangled by an arbitrarily small amount and the transformation still requires $r$ rounds. This demonstrates a type of discontinuity in the trade-off between entanglement and LOCC round number since $|00\rangle = lim_{\lambda' \to 0} |\Phi_{\lambda'}\rangle$ can be trivially obtained in zero rounds of LOCC. Such a phenomenon is reminiscent of the entanglement/round number trade-off demonstrated in ref. [20].

It is also noteworthy that the classical notion of common information played an essential role in our line of argumentation. Being able to unify the classical and quantum problems in this manner required the origami distributions to have special structure. Ozols et al. have previously used distributions of this sort to relate the tasks of classical and quantum key distillation[47], and it appears that distributions with this structure provide a useful starting point for investigating the similarities and differences between quantum entanglement and classical secrecy theories.

In addition, we remark that that the bipartite quantum states $\rho_{\mathbf{b}}^{(r,\lambda)}$ constructed in this paper exhibit entanglement reversibility in the asymptotic sense[35,49–51]. That is, the entanglement cost of generating $\rho_{\mathbf{b}}^{(r,\lambda)}$ by LOCC is equal to the amount of entanglement that can be distilled from $\rho_{\mathbf{b}}^{(r,\lambda)}$, which is $h(\lambda)$. A very simple protocol for generating $\rho_{\mathbf{b}}^{(r,\lambda)}$ at entanglement rate $h(\lambda)$ involves Alice and Bob converting $Nh(\lambda)$ copies of $|\Phi_{1/2}\rangle$ into $N$ copies of $|\Phi_\lambda\rangle$. On each of these copies Alice and Bob then choose a random joint permutation consistent with the block structure of $\mathbf{b}^{(r,\lambda)}$: $|\Phi_\lambda\rangle \to |\psi_z^{(r,\lambda)}\rangle$. Averaging over these permutation generates the state $\rho_{\mathbf{b}}^{(r,\lambda)}$. Our results show that the general structure of states possessing entanglement reversibility can be highly complex. Whether the entanglement distillation rate of $h(\lambda)$ can still be achieved for $\rho_{\mathbf{b}}^{(r,\lambda)}$ in the asymptotic sense using fewer than $r$ rounds of LOCC is an interesting question. We strongly conjecture that this is not possible, but we offer no definitive proof.

Another natural question to consider is the greatest success probability for achieving the transformations $\rho_{\mathbf{b}}^{(r,\lambda)} \to |\Phi_\lambda\rangle$ and $\mathbf{b}^{(r,\lambda)} \to \Phi_\lambda$ using $r-1$ rounds of LOCC and LOPC, respectively. The structure of the $\mathbf{b}^{(r,\lambda)}$ suggests that in both cases the success probability of any $(r-1)$-round protocol should be no greater than 1/2. In fact, it is not difficult to construct an $(r-1)$-round protocol that exactly attains the success probability 1/2. We can prove that this indeed is optimal for the classical case, but we are no longer able to easily map this bound to the quantum setting like we have done in this paper. The main reason is that monotonicity of the Schmidt/secrecy rank is no longer required in the transformation. Therefore, the unified analysis of the quantum and classical scenarios pursued in this paper no longer holds. We suspect that an LOCC/LOPC equivalence can still be established

by using tools other than the Schmidt/secrecy rank. This is left for future work.

## Methods

**Properties of the origami distributions in both classical and quantum settings.** The origami distributions belong to the more general class of "unambiguous distributions" introduced in ref. [47] since the value of one variable can always be determined from the values of the other two. The structures of unambiguous distributions lend themselves nicely to unifying LOPC and LOCC protocols via the embedding of Eq. (9). In fact, the full proof of Theorem 1, as carried out in the Supplementary Note 2, makes use of this unified structure and involves an argument that applies to both the quantum and classical problems. To compactly represent and analyze the structure of the origami distributions, we invoke in Supplmentary Note 1 the notion of common information between random variables, as proposed by Gács and Körner[52]. A slight strengthening of Theorem 1 is also presented in the Supplementary Note 3, where we relax the requirement of perfect transformation and consider $\varepsilon$-approximate transformations. Our argument involves deriving an upper bound on the $r$-round communication complexity for a general LOPC transformation of variables, a result which may be of independent interest elsewhere.

**Theorem 1 proof sketch.** The connection between the LOPC and LOCC frameworks can be seen in the basic transformation depicted by Fig. 1. The LOPC transformation $\mathbf{b}^{(1,\lambda)} \to \Phi_\lambda'$ is made coherent by replacing Bob's announcement $M$ (where $M = 0$ if $Y \in \{0, 1\}$ and $M = 1$ if $Y \in \{2, 3\}$) with the projective measurement $\{P_0, P_1\}$ (where $P_0 = |0\rangle\langle0| + |1\rangle\langle1|$ and $P_1 = |2\rangle\langle2| + |3\rangle\langle3|$) and an announcement of the measurement outcome. The result of this LOCC protocol is the transformation $\rho_{\mathbf{b}}^{(1,\lambda)} \to |\Phi_{\lambda'}\rangle$. It is then relatively easy to see that the transformations of Theorem 1 are achievable in $r$ rounds by repeating these basic LOPC/LOCC protocols to $r$ "layers" of $\mathbf{b}^{(1,\lambda)}$ with Alice and Bob alternating in the communication. The difficulty comes in showing that no fewer than $r$ rounds will succeed in accomplishing the transformations.

Proving the round-number lower bound of Theorem 1 is based on the theory of operational monotones. An LOPC/LOCC monotone is any function that cannot be increased under LOPC/LOCC processing[44,53]. While the use of LOCC monotones is a standard technique for proving impossibility results in entanglement theory, the analogous theory of LOPC monotones has received far less development and application. One of the most basic LOCC monotones is the Schmidt rank of a quantum pure state. Recall that the Schmidt rank of a bipartite pure state $|\psi\rangle^{AB}$, denoted by $Srk(|\psi\rangle)$, is equivalent to the ranks of the reduced density matrices $\rho^A = tr_B|\psi\rangle\langle\psi|$ and $\rho^B = tr_A|\psi\rangle\langle\psi|$. Unlike most LOCC monotones, the Schmidt rank has an even stronger property that it cannot be increased under LOCC even with some nonzero probability, regardless of how small this probability may be[29]. Such a monotone is called a stochastic LOCC (SLOCC) monotone.

Since $\rho_{\mathbf{b}}^{(r,\lambda)}$ is a mixture of bipartite pure states $|\psi_z^{(r,\lambda)}\rangle$, the transformation $\rho_{\mathbf{b}}^{(r,\lambda)} \to |\Phi_{\lambda'}\rangle$ requires that protocols transform $|\psi_z^{(r,\lambda)}\rangle$ into $|\Phi_{\lambda'}\rangle$ for every $z$. The origami distributions and their embeddings $|\mathbf{b}^{(r,\lambda)}\rangle$ are designed in such a way that obtaining $|\Phi_{\lambda'}\rangle$ from $|\psi_z^{(r,\lambda)}\rangle$ in $r-1$ rounds will necessarily cause the Schmidt of rank $|\psi_{z'}^{(r,\lambda)}\rangle$ to decrease for some other $z' \neq z$. Since $|\psi_{z'}^{(r,\lambda)}\rangle$ and $|\Phi_{\lambda'}\rangle$ both have a Schmidt rank of 2, a decrease in the rank of the former will make it impossible to obtain the latter. Hence the transformation $\rho_{\mathbf{b}}^{(r,\lambda)} \to |\Phi_{\lambda'}\rangle$ is impossible in $r-1$ rounds as there will always be at least one failure branch. The Schmidt rank can only be preserved along all branches if the protocol is carried out for $r$ total rounds.

One can see the general idea of this argument in greater detail by examining $\mathbf{b}^{(4,\lambda)}$ (see Eq. (8)). Suppose it is Bob who is making the first measurement and let this be described by Kraus operators $\{B_k\}_k$. Up to renormalization, the $k^{th}$ post-measurement state of each $|\psi_z^{(4,\lambda)}\rangle$ will have the form $(\mathbb{I} \otimes B_k)|\psi_z^{(4,\lambda)}\rangle$. In order for this state to ultimately reach $|\Phi_{\lambda'}\rangle$, we must have that for every $z$ either $(\mathbb{I} \otimes B_k)|\psi_z^{(4,\lambda)}\rangle$ has Schmidt rank two or $(\mathbb{I} \otimes B_k)|\psi_z^{(4,\lambda)}\rangle = 0$. Inspection of $\mathbf{b}^{(4,\lambda)}$ shows that if $(\mathbb{I} \otimes B_k)|\psi_z^{(4,\lambda)}\rangle = 0$ for some $z$ (meaning that two rows of $\mathbf{b}^{(4,\lambda)}$ have been eliminated in Eq. (8)), then $(\mathbb{I} \otimes B_k)|\psi_z^{(4,\lambda)}\rangle = 0$ for all $z$. The reason is that if a value of $z$ lies in a row being eliminated, then the Schmidt rank of $(\mathbb{I} \otimes B_k)|\psi_z^{(4,\lambda)}\rangle$ cannot be two, and therefore it must be that $(\mathbb{I} \otimes B_k)|\psi_z^{(4,\lambda)}\rangle = 0$. Therefore, if Bob is the first measuring party on $|\psi_z^{(4,\lambda)}\rangle$, each nonzero post-measurement state is SLOCC equivalent to $|\psi_z^{(4,\lambda)}\rangle$, meaning it is related to $|\psi_z^{(4,\lambda)}\rangle$ by an invertible SLOCC transformation[54]. If Alice is the first measure, a similar argument shows that her actions are also sharply limited. However, unlike Bob, she can eliminate either the right or left sub-block of $\mathbf{b}^{(4,\lambda)}$ without violating the Schmidt rank constraint. In the latter case, what remains is some state SLOCC equivalent to $\rho_{\mathbf{b}}^{(3,\lambda)}$. Thus, any LOCC protocol transforming $\rho_{\mathbf{b}}^{(4,\lambda)} \to |\Phi_{\lambda'}\rangle$ must generate a mixture $\sigma^{AB}$ that is SLOCC equivalent to $\rho_{\mathbf{b}}^{(3,\lambda)}$ along every branch in the

protocol, and it will require at least one round to do so with Alice being the measuring party.

A careful inductive argument then allows one to establish the following general result: Any LOCC protocol transforming $\rho_{\mathbf{b}}^{(r,\lambda)} \rightarrow |\Phi_{\lambda'}\rangle$ must generate a state $\sigma^{AB}$ that is SLOCC equivalent to $\rho_{\mathbf{b}}^{(1,\lambda)}$ along every branch in the protocol, and it will require at least $r - 1$ rounds to do so. From this, it follows that the transformation $\rho_{\mathbf{b}}^{(r,\lambda)} \rightarrow |\Phi_{\lambda'}\rangle$ is possible in $r - 1$ rounds only if there exists a state $\sigma^{AB}$ that is SLOCC equivalent to $\rho_{\mathbf{b}}^{(1,\lambda)}$ and which can be transformed into $|\Phi_{\lambda'}\rangle$ via local completely positive trace-preserving (CPTP) maps by Alice and Bob. The last condition holds because after $r - 1$ rounds, Alice and Bob cannot communicate further and thus they must perform local CPTP maps on $\sigma^{AB}$ in order to obtain $|\Phi_{\lambda}\rangle$. It is not difficult to show that no such $\sigma^{AB}$ exists and consequently the transformation $\rho_{\mathbf{b}}^{(r,\lambda)} \rightarrow |\Phi_{\lambda'}\rangle$ is not possible in $r - 1$ rounds.

The crucial piece in this argument is that the Schmidt rank is an SLOCC monotone, and therefore every local action must either eliminate $\left|\psi_z^{(r,\lambda)}\right\rangle$ or leave Schmidt rank unchanged. We introduce below a classical analog of the Schmidt rank called the secrecy rank that is similarly a stochastic LOPC (SLOPC) monotone; i.e., it cannot be increased even probabilistically under LOPC operations. With this monotonicity, it follows analogously to the quantum case that $\mathbf{b}^{(r,\lambda)} \rightarrow \Phi_{\lambda'}$ is not possible in fewer than $r$ rounds of communication.

**Data availability**. The authors declare that all the data supporting the findings of this study are available within the paper and its Supplementary Information Files.

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

### Acknowledgements

We thank Benjamin Fortescue for fruitful discussions on multi-round key distillation. E.C. is supported by the National Science Foundation (NSF) Early CAREER Award No. 1352326. M.H. is supported by an ARC Future Fellowship under Grant FT140100574.

### Author contributions

In this work M.-H.H. contributed most heavily to the construction of the origami distributions and E.C. contributed most heavily to the proofs.

### Additional information

**Competing interests:** The authors declare no competing financial interests.

