## [Peer Review File · Nature Communications]

Reviewers' comments:

Reviewer #1 (Remarks to the Author):

This paper studies the number of rounds of classical communication are needed to locally transform mixed-state entanglement into pure-state entanglement, or similarly, distilling insecure classical correlations into secret shared probability distributions. The natural classes of operations in these two scenarios are Local Operations and Classical Communication (LOCC), and Local Operations and Public Communication (LOPC).

A recursive construction of tripartite probability distributions and associated tripartite quantum states (obtained by embedding these probability distributions into quantum states) are constructed, such that the number of LOCC/LOPC rounds required to achieve the transformation is equal to the number of recursion steps in the construction. This is achieved by a novel recursive construction of "origami states" that involves the use of classical Gacs-Korner common information.

There is a restriction that the transformation needs to reproduce the target state *exactly*. This restriction is unnatural but it significantly helps with proving hardness. It's a bit like the exact vs approximate boson sampling -- dealing with the exact case is much easier while the approximate case (which is also practically more relevant) is way harder.

In my opinion, the current manuscript seems a bit confusing and does not convey all important messages in the following perspective:

- when introducing the "origami states", it is hard to get any intuition behind the math. It is very intuitive even after reading the supplementary materials. In particular, what conditions are important for constructing such examples? can you construct more examples or why are the current "origami states" special?
- the method part seems to be very sketchy and it is hard to see and judge the true technical novelty.
- while I do appreciate the connection between the classical secrecy rank and the Schmidt rank, I have a hard time in figuring out how this part of results fit in the main story. Thus, I doubt whether that is a good topic to put in the main text.

I think the manuscript can benefit from addressing the above issues as well as fixing a few typos.

Reviewer #2 (Remarks to the Author):

The paper describes a classical and a quantum distributed task and shows that in both cases at least r rounds of communication is needed, as well as local operations, in order to achieve it. More precisely, the setting is the following: two parties, Alice and Bob, start with some correlated random variables or a mixture of bipartite entangled states and the goal is by performing local operations and public communication, to transform these states into secret key or pure entanglement. The question is how many rounds of communication the two parties need in order to perform such transformations. The authors provide a family of classical correlations and of mixed entangled states, for which they prove that at least r rounds of communication is necessary, in the sense that any protocol with at most $(r-1)$ rounds has probability of success at most $\frac{1}{2}$ (this is only proven for the classical case and conjectured for the quantum case).

The description of previous work is not satisfactory. For example, the authors mention the model of communication complexity and claim that the "the exact manner and extent to which rounds of communication help resource manipulation is largely unknown". This is not really the case. There are results both in the classical and quantum setting that show that there are tasks that need at least r rounds of communication while any protocol that uses at most $(r-1)$ rounds or r rounds with

the wrong player starting the interaction, must be exponentially less efficient than the r round protocol. See for example the works on the Pointer Jumping function or the work "Interaction in Quantum Communication and the Complexity of Set Disjointness" by Klauck, Nayak, Ta-Shma, Zuckerman in the quantum setting.

Moreover, many of the components of the proof of the paper seem to have been used before in the literature, and the authors do not explain where exactly is the novelty in this work. For example, the classical secrecy rank and its monotonicity is a simple derivation once the known results about the Schmidt rank in the quantum setting are taken into account. In addition, the origami distribution that the authors define resemble the ones used in previous work by Ozols et al.

In conclusion, while this is an interesting result for the specific community that works on this subfield of quantum information, I do not find that the paper provides a real breakthrough nor that is of wide-enough interest to justify publication in Nature Communications.

More specific comments:

- The description of the origami distribution can be largely improved. The ranges of the variables are not defined and equation (4) is hard to parse. Consider providing the concrete probability values for the base case.
- Methods: I would prefer to see some more details about the proof of Theorem 1 in the main text, rather than the long discussion on the classical secrecy rank, which is straightforward.
- One would expect a bound on the probability of success for any quantum $(r-1)$ -round protocol.
- Last paragraph on page 2: No need to self-describe your proofs as highly non-trivial.

Reviewer #3 (Remarks to the Author):

The authors answer a longstanding open question in the theory of LOCC state transformation: whether each level of the hierarchy formed by restricting the number of rounds of interaction in LOCC protocols is a strict hierarchy. Said otherwise, is it true that for each r , the set of transformations achievable with r rounds of LOCC interaction is strictly contained in the set of transformations achievable with $r+1$ rounds of interaction. Such a strict inclusion was known to hold for some particular values of r , e.g. $r=1$. The authors settle the question for all r , explicitly constructing classes of transformations that they prove to be achievable with $r+1$ rounds of interaction but not with only r rounds.

Their work builds on recent results further elucidating the link between entanglement and secret classical correlations, as well as recent results by the same authors (and a further collaborator) elucidating the structure of distillable secret correlations.

This work helps to further elucidate the complex structure of quantum entanglement (and secret classical correlations) and will be of interest to people studying entanglement theory, definitely being one of the most significant recent papers on the topic.

Strengths: The paper studies and answer a fundamental question about the structure of LOCC protocols, and more generally it deepens our understanding of the bipartite manipulation of entanglement and of quantum information processing. It is clearly written.

Weaknesses: The states for which they can prove their round separation are quite contrived and not necessarily of the type that might naturally arise in quantum information processing.

Notwithstanding the above weaknesses, this paper makes a significant contribution to our understanding of the structure of bipartite entanglement by answering a longstanding open

question and I would recommend acceptance.

Further comments:

-The fact that the secrecy rank is an LOPC monotone is interesting and definitely worthy of mention in the main body of the text, but the proof might not be of as wide interest and might be better left as supplemental material.

-It might be worth to mention in the introduction the original work on entanglement purification: Phys.Rev.Lett.76:722-725,1996

-Another result that might be worthy of mention when discussing the possibility of round compression is the result that efficient quantum interactive proof systems with arbitrary interaction can be reduced to 3 rounds of interaction.

-In the proof of the monotonicity of secrecy rank, eq. (13), in the second inequality, would not it be more natural to state the minimization over $X-ZMW-Y$ (i.e. not YM at the end)?

-In the supplemental material, maybe mention that H and I are Shannon entropy and mutual information, respectively.

-Many typos to correct, e.g., about rather than above on p5, they rather than there on p7, Theorem number missing on p9, etc.

Dear Editors and Reviewers,

We sincerely thank the three reviewers for their detailed comments and constructive evaluations. We are pleased to see that the overall significance of our results is generally appreciated among the reviewers. In particular, Reviewer #3 concludes that

The paper studies and answer a fundamental question about the structure of LOCC protocols, and more generally it deepens our understanding of the bipartite manipulation of entanglement and of quantum information processing...[It] makes a significant contribution to our understanding of the structure of bipartite entanglement by answering a longstanding open question and I would recommend acceptance.

We would like to resubmit our manuscript for publication in *Nature Communications*. To address the primary criticisms of the reviewers and to make our submission even stronger, we have made the following three main changes, in addition to smaller changes described below.

- (1) We have clarified in the introduction how the notion of “LOPC/LOCC round complexity” studied in this paper is distinct from the more well-known “ r -round communication complexity.” LOPC/LOCC round complexity quantifies a lower bound on the number of communication exchanges needed to perform a given task, regardless of the overall communication amount – even if the total communication is exponential in the input size. This type of round complexity is a fundamental concept in the LOPC/LOCC framework where rounds of interaction cannot always be exchanged for bits of public communication.
- (2) We improved the presentation of the origami distributions and the discussion of their properties. This is done in the “Results” section where we describe the structure of distribution $\mathbf{b}^{(1,\lambda)}$ in greater detail and provide a diagram depicting a one-way distillation protocol. As $\mathbf{b}^{(1,\lambda)}$ is the basic building block for all the origami distributions, the reader will now have a much better understanding of how all the origami distributions are constructed.
- (3) We have largely expanded the “Methods” section to explain in more detail how we derive a lower bound on LOPC/LOCC round number. The main idea is described through an explicit analysis of $\mathbf{b}^{(4,\lambda)}$. We also emphasized the importance and novelty of the secrecy rank in proving our main result. Our subsection on the secrecy rank fits more cohesively now and its analogy to the Schmidt rank further substantiates the unified picture of the LOPC/LOCC that we are drawing throughout the paper. Following the advice of Reviewer #3, we have relocated the proof of Theorem 2 into the Supplemental Material.

These changes have been highlighted in maroon in the new manuscript.

We now respond to the specific comments/criticisms of each reviewer.

Reviewer #1:

Original Report:

This paper studies the number of rounds of classical communication are needed to locally transform mixed-state entanglement into pure-state entanglement, or similarly, distilling insecure classical correlations into secret shared probability distributions. The natural classes of operations in these two scenarios are Local Operations and Classical Communication (LOCC), and Local Operations and Public Communication (LOPC).

A recursive construction of tripartite probability distributions and associated tripartite quantum states (obtained by embedding these probability distributions into quantum states) are constructed, such that the number of LOCC/LOPC rounds required to achieve the transformation is equal to the number of recursion steps in the construction. This is achieved by a novel recursive construction of "origami states" that involves the use of classical Gacs-Korner common information.

There is a restriction that the transformation needs to reproduce the target state *exactly*. This restriction is unnatural but it significantly helps with proving hardness. It's a bit like the exact vs approximate boson sampling – dealing with the exact case is much easier while the approximate case (which is also practically more relevant) is way harder.

In my opinion, the current manuscript seems a bit confusing and does not convey all important messages in the following perspective:

- When introducing the origami states, it is hard to get any intuition behind the math. It is very intuitive even after reading the supplementary materials. In particular, what conditions are important for constructing such examples? can you construct more examples or why are the current origami states special?
- The method part seems to be very sketchy and it is hard to see and judge the true technical novelty.
- While I do appreciate the connection between the classical secrecy rank and the Schmidt rank, I have a hard time in figuring out how this part of results fit in the main story. Thus, I doubt whether that is a good topic to put in the main text. I think the manuscript can benefit from addressing the above issues as well as fixing a few typos.

Point-by-Point Responses from Authors

When introducing the origami states, it is hard to get any intuition behind the math. It is very intuitive even after reading the supplementary materials. In particular, what conditions are important for constructing such examples? can you construct more examples or why are the current origami states special?

Reply: As noted in point (2) above, we have now provided a more thorough discussion on the intuition behind the origami distribution. These distributions are designed to have the basic property that Alice can determine the value of Bob and Eve's variables using an r -round communication protocol that does not leak any information to Eve. If the parties try to cheat and use an $(r - 1)$ -round protocol, they will necessarily compromise their security. There will undoubtedly be other distributions besides the origami distributions that have these properties either exactly or approximately. But we believe that the origami distributions are the simplest, and they can serve as a paradigm example for further research into round complexity.

The method part seems to be very sketchy and it is hard to see and judge the true technical novelty.

Reply: As described in point (3) above, we have now provided more concrete details to the proof in the "Methods" section. The essential ideas of the proof are now described for the specific example of $\mathbf{b}^{(4,\lambda)}$. A fully exhaustive argument is still provided in the supplemental material.

While I do appreciate the connection between the classical secrecy rank and the Schmidt rank, I have a hard time in figuring out how this part of results fit in the main story. Thus, I doubt whether that is a good topic to put in the main text.

Reply: The classical secrecy rank is a crucial component in the proof of Theorem 1, and it serves as new information-theoretic object in the larger research program of unifying the LOPC and LOCC frameworks. As described in point (3) above, in the revised version of the manuscript we have provided greater explanation

of how the secrecy rank is similar to the Schmidt rank and how it can therefore be used to prove the LOPC part of Theorem 1.

Reviewer #2:

Original Report:

The paper describes a classical and a quantum distributed task and shows that in both cases at least r rounds of communication is needed, as well as local operations, in order to achieve it. More precisely, the setting is the following: two parties, Alice and Bob, start with some correlated random variables or a mixture of bipartite entangled states and the goal is by performing local operations and public communication, to transform these states into secret key or pure entanglement. The question is how many rounds of communication the two parties need in order to perform such transformations. The authors provide a family of classical correlations and of mixed entangled states, for which they prove that at least r rounds of communication is necessary, in the sense that any protocol with at most $(r-1)$ rounds has probability of success at most ϵ (this is only proven for the classical case and conjectured for the quantum case).

The description of previous work is not satisfactory. For example, the authors mention the model of communication complexity and claim that the the exact manner and extent to which rounds of communication help resource manipulation is largely unknown. This is not really the case. There are results both in the classical and quantum setting that show that there are tasks that need at least r rounds of communication while any protocol that uses at most $(r-1)$ rounds or r rounds with the wrong player starting the interaction, must be exponentially less efficient than the r round protocol. See for example the works on the Pointer Jumping function or the work "Interaction in Quantum Communication and the Complexity of Set Disjointness" by Klauck, Nayak, Ta-Shma, Zuckerman in the quantum setting.

Moreover, many of the components of the proof of the paper seem to have been used before in the literature, and the authors do not explain where exactly is the novelty in this work. For example, the classical secrecy rank and its monotonicity is a simple derivation once the known results about the Schmidt rank in the quantum setting are taken into account. In addition, the origami distribution that the authors define resemble the ones used in previous work by Ozols et al.

In conclusion, while this is an interesting result for the specific community that works on this subfield of quantum information, I do not find that the paper provides a real breakthrough nor that is of wide-enough interest to justify publication in Nature Communications.

More specific comments:

- The description of the origami distribution can be largely improved. The ranges of the variables are not defined and equation (4) is hard to parse. Consider providing the concrete probability values for the base case.
- Methods: I would prefer to see some more details about the proof of Theorem 1 in the main text, rather than the long discussion on the classical secrecy rank, which is straightforward.
- One would expect a bound on the probability of success for any quantum $(r-1)$ -round protocol.
- Last paragraph on page 2: No need to self-describe your proofs as highly non-trivial.

Point-by-Point Responses from Authors

The description of previous work is not satisfactory. For example, the authors mention the model of communication complexity and claim that "the exact manner and extent to which rounds of communication help resource manipulation is largely unknown." This is not really the case. There are results both in the classical and quantum setting that show that there are tasks that need at

least r rounds of communication while any protocol that uses at most $(r-1)$ rounds or r rounds with the wrong player starting the interaction, must be exponentially less efficient than the r round protocol. See for example the works on the Pointer Jumping function or the work “Interaction in Quantum Communication and the Complexity of Set Disjointness” by Klauck, Nayak, Ta-Shma, Zuckerman in the quantum setting.

Reply: We agree that our previous summary of prior work was not sufficient and potentially misrepresentative. As described in point (1) above, our revised version describes more clearly how our work differs from previous work on interactive protocols such as the “Pointer Jumping” problem. We still stand by the claim that little is known about LOPC/LOCC round complexity. This is a different topic than r -round communication complexity in which the latter asks how much communication is needed to perform a particular task in r -rounds of communication. In problems like pointer jumping or set disjointness there will always be a finite lower bound to the 1-round communication complexity since the parties can just broadcast all their local information to the other parties. However, in the LOPC/LOCC frameworks there are restrictions on the communication, and certain tasks become impossible in r rounds even with an unbounded amount of communication (such as generating secret key using LOPC or generating entanglement using LOCC). For this reason, we feel the round complexity studied in this paper is fundamentally distinct from most previous studies on interactive communication complexity.

The origami distribution that the authors define resemble the ones used in previous work by Ozols et al.

Reply: Indeed the origami distributions belong to the set of “unambiguous distributions” studied by Ozols et al.. We have added a reference to their paper.

The description of the origami distribution can be largely improved. The ranges of the variables are not defined and equation (4) is hard to parse. Consider providing the concrete probability values for the base case.

Reply: We have made these improvements (see point (2) above).

Methods: I would prefer to see some more details about the proof of Theorem 1 in the main text, rather than the long discussion on the classical secrecy rank, which is straightforward.

Reply: We have made these improvements (see point (3) above).

One would expect a bound on the probability of success for any quantum $(r - 1)$ -round protocol.

Reply: In fact as described in the concluding paragraph of the paper, we conjecture a lower bound of $1/2$ on the success probability for $(r - 1)$ -round protocols. However, we are currently unable to supply a rigorous proof for this fact that applies to both the classical and quantum cases.

Last paragraph on page 2: No need to self-describe your proofs as highly non-trivial.

Reply: We have removed this description.

Reviewer #3:

Original Report:

The authors answer a longstanding open question in the theory of LOCC state transformation: whether each level of the hierarchy formed by restricting the number of rounds of interaction in LOCC protocols is a strict hierarchy. Said otherwise, is it true that for each r , the set of

transformations achievable with r rounds of LOCC interaction is strictly contained in the set of transformations achievable with $r+1$ rounds of interaction. Such a strict inclusion was known to hold for some particular values of r , e.g. $r=1$. The authors settle the question for all r , explicitly constructing classes of transformations that they prove to be achievable with $r+1$ rounds of interaction but not with only r rounds.

Their work builds on recent results further elucidating the link between entanglement and secret classical correlations, as well as recent results by the same authors (and a further collaborator) elucidating the structure of distillable secret correlations.

This work helps to further elucidate the complex structure of quantum entanglement (and secret classical correlations) and will be of interest to people studying entanglement theory, definitely being one of the most significant recent papers on the topic.

Strengths: The paper studies and answer a fundamental question about the structure of LOCC protocols, and more generally it deepens our understanding of the bipartite manipulation of entanglement and of quantum information processing. It is clearly written.

Weaknesses: The states for which they can prove their round separation are quite contrived and not necessarily of the type that might naturally arise in quantum information processing.

Notwithstanding the above weaknesses, this paper makes a significant contribution to our understanding of the structure of bipartite entanglement by answering a longstanding open question and I would recommend acceptance.

Further comments:

- The fact that the secrecy rank is an LOPC monotone is interesting and definitely worthy of mention in the main body of the text, but the proof might not be of as wide interest and might be better left as supplemental material.
- It might be worth to mention in the introduction the original work on entanglement purification: Phys.Rev.Lett.76:722-725,1996
- Another result that might be worthy of mention when discussing the possibility of round compression is the result that efficient quantum interactive proof systems with arbitrary interaction can be reduced to 3 rounds of interaction.
- In the proof of the monotonicity of secrecy rank, eq. (13), in the second inequality, would not it be more natural to state the minimization over X -ZMW- Y (i.e. not YM at the end)?
- In the supplemental material, maybe mention that H and I are Shannon entropy and mutual information, respectively.
- Many typos to correct, e.g., about rather than above on p5, they rather than there on p7, Theorem number missing on p9, etc.

Point-by-Point Responses from Authors

The states for which they can prove their round separation are quite contrived and not necessarily of the type that might naturally arise in quantum information processing.

Reply: We agree that the origami distributions have been tailored to ease in the mathematical analysis. However, the overall objective of this paper is to provide a proof of principle on the type of complexity that can emerge in the optimal processing of classical/quantum resources via interactive LOPC/LOCC, something previously unknown within the quantum information community. We believe that the origami distributions are not unique in having this property.

The fact that the secrecy rank is an LOPC monotone is interesting and definitely worthy of mention in the main body of the text, but the proof might not be of as wide interest and might be better left as supplemental material.

Reply: We have streamlined the subsection on secrecy rank and placed the proof of Theorem 2 into the Supplemental Material as suggested.

It might be worth to mention in the introduction the original work on entanglement purification: Phys.Rev.Lett.76:722-725,1996

Reply: We have added this reference to the introduction.

Another result that might be worthy of mention when discussing the possibility of round compression is the result that efficient quantum interactive proof systems with arbitrary interaction can be reduced to 3 rounds of interaction.

Reply: We thank the reviewer for making this suggestion, and ultimately the suggested reference shows that there is a precedence for round compression in interactive quantum communication protocols. We have included a reference to this paper.

In the proof of the monotonicity of secrecy rank, eq. (13), in the second inequality, would not it be more natural to state the minimization over $X - ZMW - Y$ (i.e. not YM at the end)?

Reply: The minimization over $X - ZMW - Y$ is equivalent to $X - ZMW - YM$. The reason we choose to put the latter is to indicate that it is Alice who generates the message M and not Bob. We think this might make the argument conceptually easier to follow even though it is mathematically equivalent to the reviewer's suggestion. (Since this is public communication, ultimately it does not matter who generates the message for the purpose of computing the minimization).

In the supplemental material, maybe mention that H and I are Shannon entropy and mutual information, respectively.

Reply: We have added this.

Many typos to correct, e.g., about rather than above on p5, they rather than there on p7, Theorem number missing on p9, etc.

Reply: We have fixed these as well as others.

Thank you for reconsidering our manuscript. With these changes, we believe it will make a perfect fit in *Nature Communications*.

Sincerely,

Eric Chitambar and Min-Hsiu Hsieh

REVIEWERS' COMMENTS:

Reviewer #2 (Remarks to the Author):

The paper is considerably improved and has taken into account the reviewers' comments in a satisfactory manner.